# Iterative reconstruction artefact removal using null-space networks

**Author(s) names withheld**                                           EMAIL(S) WITHHELD

## Abstract

Incorporation of resolution modelling (RM) into iterative reconstruction produces Gibbs ringing artefacts which adversely affect clinically used metrics such as $SUV_{max}$. We propose the use of a null space network as a regulariser to compensate for these artefacts without introducing bias.

**Keywords:** Null net, data consistency, image reconstruction, PET

## 1. Introduction

Maximum likelihood expectation maximisation (MLEM) is a versatile iterative algorithm frequently used in clinical PET image reconstruction. Incorporating resolution modelling (RM) into MLEM results in resolution recovery but also introduces Gibbs ringing artefacts. Since these artefacts affect clinically important metrics such as maximum standardised uptake values ($SUV_{max}$), there is debate as to whether RM should be used at all (Alessio, Rahmim, and Orton 2013).

Ringing artefacts are caused by a sharp drop in recovery of high frequency components. The unrecoverable data exists in the null space of the imaging system. We propose the use of a deep null space network (Schwab, Antholzer, and Haltmeier 2019a) to fill in this null space, thereby removing artefacts in a data-consistent manner. Null nets have recently been applied to filtered backprojection (FBP) of sparse photoacoustic tomography (PAT) (Li et al. 2019) and undersampled Radon transforms (Schwab, Antholzer, and Haltmeier 2019b).

## 2. Methods

Data consistency (more specifically, model consistency) is enforced by incorporating MLEM reconstruction into the training of the proposed network. Once trained, the network can be applied as a post-reconstruction step to test data. Due to the presence of noise in real data, full consistency is not desirable. Instead, consistency is only enforced here with respect to resolution degradation and recovery.

**Data**: 20 *BrainWeb* phantoms are modified to have PET-like intensities and spherical lesions of diameters ranging from 2 to 15 mm as described in da Costa-Luis (2019). The *BigBrain* FDG PET phantom (Belzunce 2018) was also registered with the *BrainWeb* data and Poisson noise added corresponding to 30 M counts. Resolution degradation is achieved by Gaussian smoothing with 5 mm full width at half maximum (FWHM). This data is then

reconstructed using 300 MLEM iterations . In the case of purely performing resolution recovery, MLEM is equivalent to the Richardson-Lucy (RL) algorithm (Appendix A).

**Training**: 18 noise-free *BrainWeb* phantoms are used for training, and 2 for validation. The *BigBrain* phantom is used for testing. The task is to transform RL reconstructions into artefact-free ground truth predictions in a data-consistent manner. Consistency means the forward model (Gaussian smoothing) should produce identical results whether applied to the input (RL reconstructions) or to the predictions.

The proposed post-processing step is:

$$P(\hat{\boldsymbol{x}}) = \hat{\boldsymbol{x}} + N(\hat{\boldsymbol{x}}) - \mathcal{R}_k(G(N(\hat{\boldsymbol{x}}))), \tag{1}$$

where $\mathcal{R}_k$ represents $k$ MLEM (Richardson-Lucy) iterations,
  $G$  applies the forward model (Gaussian smoothing),
  $N$  is a 4-layer convolutional network, and
  $\hat{\boldsymbol{x}}$  is reconstructed (using $\mathcal{R}_k$) data.

$N$ consists of 4 fully 3D convolutional layers with unit stride, width 3, and zero-padding; each of which are followed by a *ReLU* activation function. The first 3 layers have 32 kernels, while the final layer has 1. The normalised root mean square error (NRMSE) is used as the loss function:

$$L(P; \boldsymbol{y}) = \sum_n \sqrt{\|\boldsymbol{y}_n - P(\mathcal{R}_k(G(\boldsymbol{y}_n)))\|^2 / \|\boldsymbol{y}_n\|^2}, \tag{2}$$

where $\boldsymbol{y}_n$ is the $n^{\text{th}}$ ground truth volume.

The Adam optimiser is used with learning rate $10^{-3}$ and convolutional kernel regularisation with weighting factor $10^{-4}$. Once trained, $P$ satisfies $G(P(\hat{\boldsymbol{x}})) = G(\hat{\boldsymbol{x}})$.

For comparison, a network $C$ (with the same architecture as $N$) is trained on the same data. NRMSE is also used as the loss for $C$, i.e. $L(C; \boldsymbol{y})$. The network $C$ thus performs the same post-processing task as $P$, but without any consistency constraint.

## 3. Results

$P$ has better generalisability than $C$ due to the consistency constraint. Similar qualitative results (not shown) were observed for real data. Future work will consider incorporating the PET forward and backprojectors into the reconstruction.

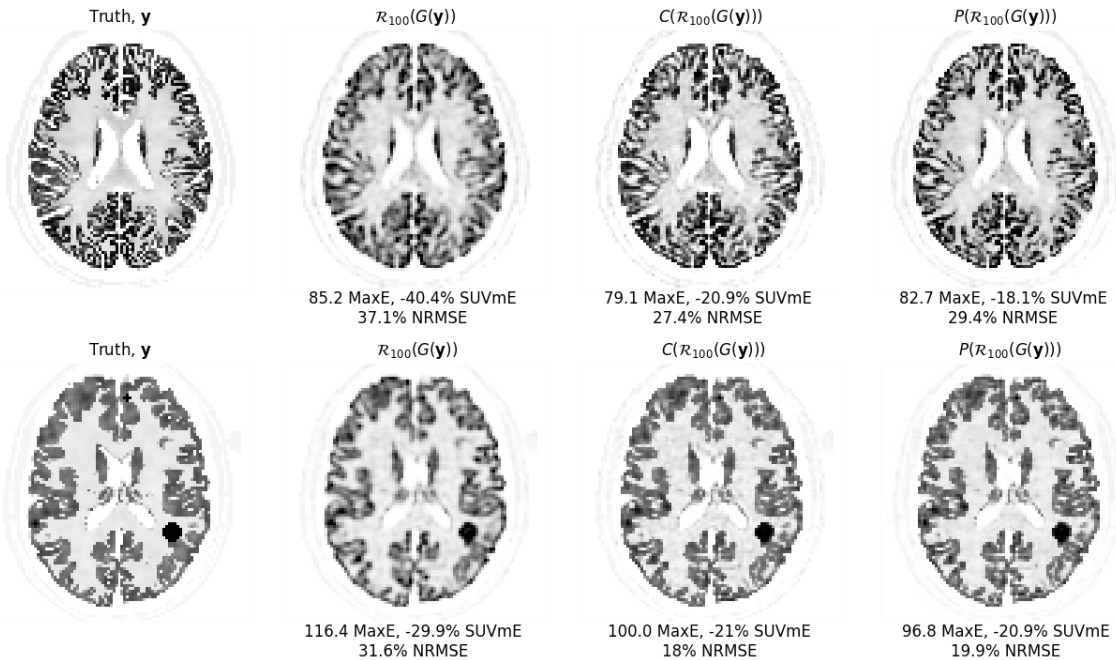

Figure 1: Results for test (top row) data (not used during training). Training data is also shown in the bottom row.

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

## Appendix A. Richardson-Lucy Iterative Reconstruction

---

**Algorithm 1:** Richardson-Lucy (RL)

---

**Input:** $m$: blurred image, $\sigma$: Gaussian blur parameter, $k$: number of iterations
**Output:** $y$: reconstructed image
$y \leftarrow \text{ones}(\text{shape}(m))$
**for** $i \leftarrow 1$ **to** $k$ **do**
$\quad \mid \quad y \leftarrow y \times G_\sigma(m \div G_\sigma(y))$          `// G is a Gaussian blur function`
**end**

---

