# OpenReview forum: "Iterative reconstruction artefact removal using null-space networks"
_MIDL.io/2020/Conference — Submitted to MIDL 2020_

### Official Review · AnonReviewer3 · 2020-02-27
**Iterative reconstruction artefact removal using null-space networks**

**Rating:** 2
**Confidence:** 2

**Review:**

The authors present in their submitted short paper manuscript an improved approach to recover/interpolate resolution (resolution modelling, RM) in PET imaging after maximum likelihood expectation maximisation (MLEM) reconstruction. The main idea is to regularize RM by a null space network to reduce reconstruction related Gibbs ringing artefacts. Overall, the idea of a regularizer network is timely and worth investigation. Digital phantom data (MRI phantom) were transformed to PET-like phantoms. In line to these training data, the authors demonstrated the potential of their approach on a digital test phantom. It is claimed that similar results could be achieved in real PET data, however, no reconstruction results were shown nor quantitatively evalulated.
Main critics (below) are formal and address the missing links/context to other work and current scientific research.

Gain:
- proof of concept and introduction of a regularized approach which also seems to generalize well
- training completely on digital phantom data (synthesized from MRI data) overcomes problems and limitations with data availability and still seems to perform well
- translation to real data expected (as the authors claim)

Shortcoming:
- overall, the manuscript has some shortcomings in the methods section. It has to be acknowledged, that this is a short paper submission. However, a discussion is not done at all and therefore, the authors do not put their results into context or point out potential or limitations.
- Equations are given in the most compressed manner, which is acceptable, but may not allow reproduction of the shown results. In particular, because training and test data are publicly available, this could be of interest with respect to open/shared science.

---

### Official Review · AnonReviewer2 · 2020-03-09
**Data consistency approach to remove artefact from iterative reconstruction**

**Rating:** 3
**Confidence:** 3

**Review:**

The short abstract is very interesting. Data consistency is known to be a fundamental building block in many image reconstruction models. It is not mentioned in the title, but the method is applied to PET images. There was no extensive validation of the approach and it is ok in a short abstract scope, but the results in the test set seem to be ~50% worse (NRMSE) than the results on the train set, which raises some model generalisability concerns.

---

### Official Review · AnonReviewer1 · 2020-03-13
**This is a short paper for PET imaging based on null-space network to remove artefacts.**

**Rating:** 2
**Confidence:** 4

**Review:**

This paper proposed to use a null-space network to remove ringing artefacts in PET image reconstruction. Overall, the novelty seems to be limited because the major novelty of null-space network is from a reference paper. Moreover, this reconstruction network is learned and tested on phantoms. Can this learned network be extensible to real PET imaging? Whether there is way to collect and train the network on real PET image?

Overall, the novelty in methodology seems to be limited and the experimental evaluations are not convincing. I suggest to more deeply investigate along this research, with improved novelties and more extensive experiments.

---

### Official Review · AnonReviewer4 · 2020-03-14
**Short and vague**

**Rating:** 2
**Confidence:** 4

**Review:**

The method is reasonable, but the paper is very short/vague and the method not tested with real data.  Short papers are allowed to use up to 3 pages excluding references, but this paper only used a fraction of that space.  The paper could have been much stronger if it used the full space allotment.

I wish the paper would have spent more time explaining why Eq. (1) was chosen given many other possibilities, since the form of Eq. (1) is not intuitive.  Since this is a standard superresolution problem, so the lack of any comparison against other superresolution methods is an important omission.

---

### Meta-Review · Area_Chair1 · 2020-03-31
**MetaReview of Paper26 by AreaChair1**

**Rating:** 1

**Metareview:**

The paper lacks novelty and real scenario validation. Even if this is a short paper, the paper still lacks susbtantial details that are needed for the readers to understand both the theoretical and experimental contributions.

**Paper Type:**

methodological development

---

### Decision · Program_Chairs · 2020-04-11

Reject